# Non-Invasive Estimation of Central Systolic Blood Pressure by Radial Tonometry: A Simplified Approach

**DOI:** 10.3390/jpm13081244

**Published:** 2023-08-10

**Authors:** Denis Chemla, Davide Agnoletti, Mathieu Jozwiak, Yi Zhang, Athanase D. Protogerou, Sandrine Millasseau, Jacques Blacher

**Affiliations:** 1INSERM UMRS 999, Hôpital Marie Lannelongue, 92350 Le Plessis-Robinson, France; denis.chemla56@gmail.com; 2Hypertension and Cardiovascular Risk Research Center, Department of Medical and Surgical Sciences, University of Bologna, 40138 Bologna, Italy; 3Cardiovascular Internal Medicine, Heart Chest and Vascular Department, IRCCS Azienda Ospedaliero-Universitaria di Bologna, 40138 Bologna, Italy; 4Service de Médecine Intensive Réanimation CHU de Nice, 06200 Nice, France; jozwiak.m@chu-nice.fr; 5UR2CA, Unité de Recherche Clinique Côte d’Azur, Université Côte d’Azur, 06200 Nice, France; 6Research Center for Epidemiology and Biostatistics, Sorbonne Paris Cité (CRESS), Université Paris Cité, AP-HP, Diagnosis and Therapeutic Center, Hôtel-Dieu, 1, Place du Parvis Notre Dame, 75004 Paris, France; yizshcn@gmail.com (Y.Z.); jacques.blacher@aphp.fr (J.B.); 7Department of Cardiology, Shanghai Tenth People’s Hospital, Tongji University School of Medicine, Middle Yanchang Road 301, Shanghai 200072, China; 8Cardiovascular Prevention & Research Unit, Clinic & Laboratory of Pathophysiology, Department of Medicine, National and Kapodistrian University of Athens, 10679 Athens, Greece; aprotog@med.uoa.gr; 9Pulse Wave Consulting, 95320 Saint Leu La Foret, France; sandrine_millasseau@yahoo.fr

**Keywords:** central pressure, aortic pressure, arterial tonometry, central hemodynamics, mean arterial pressure, systolic blood pressure

## Abstract

Backround. Central systolic blood pressure (cSBP) provides valuable clinical and physiological information. A recent invasive study showed that cSBP can be reliably estimated from mean (MBP) and diastolic (DBP) blood pressure. In this non-invasive study, we compared cSBP calculated using a Direct Central Blood Pressure estimation (DCBP = MBP^2^/DBP) with cSBP estimated by radial tonometry. Methods. Consecutive patients referred for cardiovascular assessment and prevention were prospectively included. Using applanation tonometry with SphygmoCor device, cSBP was estimated using an inbuilt generalized transfer function derived from radial pressure waveform, which was calibrated to oscillometric brachial SBP and DBP. The time-averaged MBP was calculated from the radial pulse waveform. The minimum acceptable error (DCBP-cSBP) was set at ≤5 (mean) and ≤8 mmHg (SD). Results. We included 160 patients (58 years, 54%men). The cSBP was 123.1 ± 18.3 mmHg (range 86–181 mmHg). The (DCBP-cSBP) error was −1.4 ± 4.9 mmHg. There was a linear relationship between cSBP and DCBP (R^2^ = 0.93). Forty-seven patients (29%) had cSBP values ≥ 130 mmHg, and a DCBP value > 126 mmHg exhibited a sensitivity of 91.5% and specificity of 94.7% in discriminating this threshold (Youden index = 0.86; AUC = 0.965). Conclusions. Using the DCBP formula, radial tonometry allows for the robust estimation of cSBP without the need for a generalized transfer function. This finding may have implications for risk stratification.

## 1. Introduction

Hypertension, the leading cardiovascular risk factor, is diagnosed and managed based on brachial-cuff blood pressure (BP). There is an ongoing debate regarding whether there is a stronger association of cardiovascular clinical endpoints with the pressure in ascending aorta (central BP) compared to brachial artery (peripheral BP), and whether this has any significant clinical implications [1,2,3,4]. From a pathophysiological standpoint, it is intuitively understood that cardiovascular complications and end-organ damage due to pressure overload may be more closely related to central BP rather than peripheral BP. This is because the central BP, and not the brachial BP, directly affects the heart’s workload. The proximity of the aorta, rather that the brachial artery, to the brain and kidneys is in line with this notion. Additionally, for a given cardiac output, mean BP (MBP) is determined by the systemic vascular resistance, which is predominantly controlled by the peripheral small arteries. On the other hand, for a given stroke volume, pressure fluctuations in the arterial tree are determined by arterial compliance, which is primarily influenced by the aorta and the larger arteries [5,6].

Various waveform acquisition techniques were developed to non-invasively estimate central systolic BP (cSBP) [7,8]. Among these techniques, the SphygmoCor, which utilizes radial applanation tonometry, is the most widely used non-invasive system to estimate cSBP through an inbuilt validated generalized transfer function, able to derive central waveform from radial pressure curve. It underwent successful validation, primarily as a type-I device [9,10,11]. However, interpretation of available studies in terms of risk stratification is limited due to numerous factors, including the small sample sizes and the lack of consensus regarding the best calibration method. 

Our group recently conducted an invasive study that introduced a novel method for estimating cSBP called Direct Central Blood Pressure (DCBP) estimation. This method relies solely on peripheral MBP and diastolic blood pressure (DBP) values [12]. The DCBP is calculated using the following straightforward equation:DCBP = MBP^2^/DBP

The rationale behind this approach is based on the following facts: (i) central MBP may be reliably calculated by taking the square root of the product of central SBP and DBP (geometric mean) [13]; and (ii) MBP and DBP in peripheral large arteries undergo minimal changes compared to their central counterparts [14,15,16,17] (see also Appendix A). 

While the high accuracy and precision of DCBP have been initially documented using invasively obtained BPs [12], it is unclear how DCBP obtained from non-invasive BP measurements compares to cSBP estimated using radial applanation tonometry and the standard application of generalized transfer functions. Hence, the objective of our non-invasive tonometric study was to compare type-I SphygmoCor-derived cSBP with DCBP in a prospective study conducted according to our routine protocol [18].

## 2. Materials and Methods

The present study is a retrospective analysis of an observational prospective study that enrolled 186 consecutive patients who were recruited from the cardiovascular department at the Diagnostic and Therapeutic Center of Hotel-Dieu Hospital in Paris, France, between January and June 2010. These patients were referred for the evaluation of one or more cardio-vascular risk factors, including high-blood pressure, smoking, dyslipidemia, diabetes mellitus, and/or family history of premature cardio-vascular disease. Some patients had previously experienced clinical events. The inclusion criteria required the presence of an adequate quality of the pulse wave tonometric recording of radial arteries. Exclusion criteria encompassed patients under 18 years of age, those with acute medical conditions, atrial fibrillation, and severe heart failure (NYHA III-IV). Patients provided informed consent for additional non-invasive hemodynamic measurements and data collection during the day-hospital cardiovascular screening. The study complies with the Declaration of Helsinki. The study was registered with the French National Agency for Medicines and Health Products Safety (No. 2013-A00227-38) and was approved by the locally appointed ethics committee: the Advisory Committee for Protection of Persons in Biomedical Research.

### 2.1. Collection of Clinical and Laboratory Parameters 

Clinical and laboratory parameters were collected during the day-hospital visit using a form filled-out at inclusion. The form included age, sex, weight, and height, which were determined using a stadiometer affixed to a wall and a Tanita scale with a digital read-out. Body mass index (BMI) was calculated as weight (kg) divided by height^2^ (m^2^). Additional information gathered from the form included family history of premature cardiovascular events among first-degree relatives, personal history of dyslipidemia, hypertension, smoking, previous diseases, and use of medications. Medication information, such as an antidiabetic drugs, lipid-lowering agents, and antihypertensive drugs, was obtained from patients’ files and self-reporting. Previous cardiovascular events, including coronary heart disease, cerebrovascular disease, and peripheral vascular disease, were retrospectively assessed both by scrutinizing the patients documents and by patients interviewing. Hypertension was defined as brachial SBP ≥ 140 mmHg and/or a DBP ≥ 90 mmHg, and/or use of antihypertensive drugs, following European recommendations [19]. Diabetes mellitus was defined as a glycosylated hemoglobin (HbA1c) ≥ 6.5% and/or fasting glucose level ≥ 7.0 mmol/L and/or the use of oral hypoglycemic agents or insulin therapy. Dyslipidemia was defined as a total/high-density lipoprotein cholesterol ratio greater than five or the use of hypocholesterolemic drugs. Laboratory parameters were measured on the day of the hemodynamic measurements. These parameters included plasma glucose and glycated hemoglobin levels, total cholesterol, low-density lipoprotein, and high-density lipoprotein levels, triglyceride levels, plasma creatinine levels, and estimated glomerular filtration rate (eGFR), calculated according to the MDRD formula (Modification of Diet in Renal Disease) in units of mL/min/1.73 m^2^. A calculated eGFR of less than 60 mL/min/1.73 m^2^ indicated kidney failure. The presence of albuminuria was assessed through a 24 h urine collection, and recorded as normo-albuminuria (less than 30 mg/24 h), microalbuminuria (30–300 mg/24 h), and proteinuria (more than 300 mg/24 h).

### 2.2. Hemodynamic Parameters

Hemodynamic measurements were conducted in the supine position in the morning after an overnight fast. Brachial BP was determined using a validated oscillometric device (SCVL, Paris, France), following at least 5 min of rest in the supine position. Brachial SBP and DBP were measured in both arms using cuffs of appropriate sizes (utilizing 3 different sizes) [20,21]. After three measurements taken 1 min apart, the latter two values were averaged, and the heart rate was recorded. Following BP measurement, non-invasive arterial applanation tonometry (SphygmoCor device, v8.2, AtCor Medical, Sydney, Australia) was employed to obtain the radial pulse pressure waveform. By applying a generalized transfer function, central BP was derived from the radial waveform calibrated using SBP and DBP measured at the brachial artery level. The MBP was estimated as the time-averaged mean value of this calibrated radial pressure waveform, displayed by the Sphygmocor software (version 8.1). For comparative purposes, we also calculated MBP by using the rule of thumb at the radial artery level (MBP = DBP + 33% pulse pressure (PP = SBP − DBP). Carotid-femoral pulse wave velocity (cfPWV) was calculated using the SphygmoCor device from carotid and femoral pressure waveforms, following the standard procedure. The formula for the Direct Central Blood Pressure estimation (DCBP) was: DCBP = MBP^2^/DBP. 

### 2.3. Statistical Analysis

Statistical analysis was conducted using MedCalc11.6.0 software (MedCalc, Mariakerke, Belgium). Agreement between DCBP and cSBP was evaluated by calculating the difference (error) between the two. The accuracy (mean error) and precision (standard deviation SD) of the DCBP estimate were computed. An acceptable mean error was defined as ≤5 mmHg and an acceptable SD of the error was set at ≤8 mmHg [22]. To categorize the difference, as previously recommended [23], the rounded absolute value was divided into four bands: 0–5 mmHg, representing very accurate measurements with no clinically relevant error; 6–10 mmHg, indicating slightly inaccurate measurements; 11–15 mmHg, representing moderately inaccurate measurements; and >15 mmHg, indicating highly inaccurate measurements. The final analysis was based on how the values in these bands cumulatively fell into three zones: within 5 mmHg (representing values in the 0–5 mmHg band), within 10 mmHg (representing values in the 0–5 and 6–10 mmHg bands), and within 15 mmHg (representing values in the 0–5, 6–10 and 11–15 mmHg bands). The error was also expressed as a percentage of cSBP. Bland–Altman analysis was utilized to study the agreement between DCBP and cSBP. Correlations between variables were assessed using Pearson’s correlation coefficient. Comparisons between groups were performed using either Student’s unpaired t test or analysis of variance. Central systolic hypertension was defined as cSBP ≥ 130 mmHg. Using a DCBP threshold value ≥ 130 mmHg, the percentage of patients correctly classified in the overall population was calculated and the Chi-square test was used to assess the level of concordance in classification. Finally, ROC curve analysis was performed to test the ability of DCBP in confirming the diagnosis of central systolic hypertension. A *p* value < 0.05 was considered statistically significant. The power calculation analysis retrieved values close to 1 both for the correlation and the Chi-square analyses. 

## 3. Results

Out of the 186 patients initially enrolled for a cardiovascular checkup at the day hospital, 26 patients were excluded due to poor quality of the tonometric signal or missing blood pressure data. Therefore, the final analysis included 160 patients. Anamnestic and biological data were available for 133 of included patients. The key characteristics of the study population are presented in Table 1. 

The average age of the participants was 58 years, with 54% of them being male. Two-thirds of the patients had a higher-than-normal BMI, and 79% were receiving treatments for hypertension. Additionally, 22% of the participants had diabetes mellitus. 

The hemodynamic characteristics are listed in Table 2.

The central systolic blood pressure (cSBP) was 123.1 ± 18.3 mmHg and DCBP was 121.6 ± 18.1 mmHg. A strong linear relationship between cSBP and DCBP was observed (R^2^ = 0.93; *p* < 0.001) (Figure 1).

The DCBP-cSBP error was −1.4 ± 4.9 mmHg (Figure 2).

There was no influence of the mean on the error (*p* = 0.52). The absolute error fell within the 5 mmHg zone (0 to 5 mmHg) in 126 out of 160 patients (79%), within the 10 mmHg zone (0 to 10 mmHg) in 153 patients (96%), and within the 15 mmHg zone (0 to 15 mmHg) in 158 patients (99%). The error was −1.1 ± 3.9% cSBP (95% CI–9.1; 6.7%), and it was ≤10% cSBP in 154 patients (96.3%). 

The (DCBP-cSBP) error was positively correlated with heart rate (R^2^ = 0.43, *p* < 0.01) and it was negatively correlated with age (R^2^ = 0.15, *p* < 0.01) and cfPWV (R^2^ = 0.02, *p* = 0.048). The error was not related to brachial blood pressure (systolic, diastolic, pulse) or body height, body weight and BMI. The (DCBP-cSBP) error was −2.5 ± 4.6 mmHg in men (n = 87) and 0.2 ± 5.0 mmHg in women (n = 73) (*p* < 0.01). The (DCBP-cSBP) error was −0.2 ± 4.3 mmHg in patients aged <60 years (n = 81) and −2.8 ± 5.2 mmHg in patients aged ≥60 years (n = 79) (*p* < 0.01). The (DCBP-cSBP) error was −0.5 ± 5.2 mmHg in patients with PWV < 11.3 m/s (n = 80) and −2.4 ± 4.4 mmHg in patients with PWV ≥ 11.3 m/s (n = 80) (*p* < 0.01).

Brachial SBP was ≥140 mmHg in 51 subjects (32%) and cSBP was ≥130 mmHg in 47 subjects (29%). The DCBP showed 93% concordance in discriminating a cSBP threshold of 130 mmHg (Chi-squaretest *p* < 0.0001) (Table 3). A DCBP value > 126 mmHg exhibited a sensitivity of 91.5% and specificity of 94.7% in discriminating a cSBP threshold of 130 mmHg (Youden index = 0.86; AUC = 0.965 (95% CI 0.923 to 0.987)) (Appendix A).

The radial MBP estimated by the rule of thumb (DBP + 0.33 PP) was slightly higher than the MBP obtained through pulse waveform analysis, with a mean difference of 0.9 ± 3.6 mmHg between the two methods (Table 2 and Appendix A). When calculating the DCBP, by using the radial MBP estimated by this rule of thumb, the (DCBP-cSBP) error was 0.8 ± 9.0 mmHg (1.0 ± 7.3%) (Appendix A).

## 4. Discussion

The primary finding of our non-invasive tonometric study was that DCBP demonstrated both accuracy and precision in estimating cSBP, eliminating the need for a generalized transfer function. This could present important implications for risk stratification, particularly if errors in the cuff-blood pressure measurement used to calibrate the tonometer are minimized, ensuring the reliability of DCBP. 

In a recent proof of concept and validation study, we demonstrated the reliable estimation of invasive cSBP using the invasive MBP^2^/DBP ratio that we called DCBP, which we referred to as Direct Central Blood Pressure estimation [12]. It is important to note that the high accuracy and good precision of DCBP in that study were likely dependent upon the use of intra-arterial blood pressure measurements, which may not reflect real life conditions [11,24,25]. In our current non-invasive study, we utilized tonometry to capture the radial pressure wave, which was then calibrated to oscillometric brachial SBP and DBP. Mean blood pressure (MBP) was estimated as the time-averaged mean value of this calibrated radial pressure waveform, as provided by the tonometer software (version number 8.1). Overall, our findings confirm the previous conclusion obtained from invasive measurements, which suggests that DCBP and cSBP can be considered as interchangeable. We observed a small error between the two measures (−1.4 ± 4.9 mmHg) compared to the minimum acceptable error of ≤5 ± ≤8 mmHg, according to AAMI criteria [22]. Furthermore, the absolute error fell within the 10 mmHg range in 96% of the patients. Based on our non-invasive tonometric study, it appears that the use of a transfer function is unessential for estimating cSBP. 

When MBP was calculated solely using oscillometric blood pressures and a fixed form factor (FF) of 0.33, DCBP remained a highly accurate estimate of cSBP with a mean error of 0.8 mmHg. However, it is important to note that the precision of the cSBP estimate was moderate, as indicated by the SD of the error, which was 9 mmHg. The FF represents the fraction of pulse pressure that needs adding to DBP in order to estimate MBP. It serves as a pulse shape indicator influenced by factors such as arterial location, demographic characteristics, hemodynamics, pathologies, and methodologic factors. Our finding supports the notion that using a single FF is valuable from the viewpoint of predicting MBP average values, but it fails to capture the variations in the pressure waveform among individuals. Therefore, whenever possible, it is recommended to prioritize pressure waveform analysis over relying solely on a fixed form factor to calculate MBP [18,26]. 

DCBP is likely to possess similar strengths and limitations as cSBP estimated through the transfer function of the Sphygmocor device [7,11,15,27]. When calibrating the tonometric signal with brachial-cuff measurements, there are inherent inaccuracies compared to invasive brachial artery blood pressure recordings [15,28]. Any overestimation or underestimation of true intra-arterial blood pressure by brachial-cuff measurements will impact the accuracy of DCBP [12], as in the case with SphymoCor-derived cSBP estimation [23,25,29]. Systematic reviews and meta-analyses of invasive central validation studies of commercial devices calibrated using non-invasively measured peripheral blood pressures have reported pooled estimates of the mean error of −8.2 mmHg [24], and −5.81 mmHg [25]. This discrepancy can be attributed to the fact that regardless of the non-invasive method used (oscillometric or auscultatory), DBP tends to be overestimated while SBP tends to be underestimated compared to intra-arterial values [28]. We utilized radial calibration with cuff SBP and DBP (a type I device), which is a commonly employed method in current clinical studies. However, this approach overlooks the potential amplification of blood pressure from the brachial to the radial arteries [30], and this may underestimate cSBP. 

The limitations of our study should be acknowledged. Firstly, there are various techniques available for non-invasively estimating cSBP, including tonometry, oscillometry, and echo-tracking. Our comparative results specifically pertain to the SphygmoCor device, which is currently the most widely used non-invasive system. Therefore, the generalizability of our findings to other waveform acquisition techniques remains to be studied. Secondly, it is important to note that our results are specific to the population under investigation. The mean age of the participants was 58 years with 59% being men. The prevalence of treated hypertension was 79%, while diabetes mellitus was present in in 22%. Additionally, a substantial portion of the population (64%) has a higher-than-normal BMI. Therefore, our results may not be directly applicable to younger patients or patients with different demographic or clinical characteristics. Further research is needed to validate our findings in these specific populations. Lastly, while our study focused on the accuracy of the single DCBP calculation, it is important to highlight that most of transfer function-based methods enable the estimation of the entire pressure wave shape and provide valuable arterial indices, such as the augmentation index. These indices have clinical relevance and may provide insights into arterial function beyond cSBP estimation.

The implications of our study warrant discussion. Previous research has suggested that cSBP may have incremental value above and beyond brachial BP in diagnosis and management of patients with hypertension [2,31]. However, studies comparing the association between adverse health outcomes and brachial systolic blood pressure (SBP) versus central systolic blood pressure (cSBP) have faced limitations, as reviewed in the literature [3,11]. These limitations include the use of different technologies to estimate cSBP and the inherent constraints of sample sizes. Based on our findings, a simple radial pressure waveform analysis combined with brachial oscillometry, resulting in DCBP, provides a highly accurate and precise estimation of cSBP without the need for a transfer function. Moreover, when a FF of 0.33 is employed instead of radial pulse waveform to estimate MBP, DCBP maintains its high accuracy, with a mean error of less than 1 mmHg. This alternative approach, utilizing brachial oscillometry alone and a fixed FF, can be particularly advantageous in medical facilities that do not have access to tonometry equipment. DCBP can be calculated on existing BP data, thus providing accurate cSBP estimations, and thereby expanding the potential inclusion of patients in research studies or clinical settings. An important issue to consider is that few studies analyzing the reference values of cSBP are available. The lack of large-scale worldwide accepted reference values for cSBP should be addressed in future research. At the same time, normative values for DCBP should be calculated or derived to implement the formula in population settings. Finally, our study indicates that DCBP may be useful in diagnosing central systolic hypertension, and further studies are needed to confirm this finding and to document the potential prognostic association of DCBP with cardiovascular risk. The most relevant clinical application of DCBP would be to address the CV risk in healthy subjects. Indeed, the detection of early vascular ageing, before any classical CV risk factor is established, would allow for better risk stratification and, consequently, a more rigorous follow-up.

## 5. Conclusions

In conclusion, our non-invasive study provides compelling evidence that DCBP (MBP^2^/DBP) is a highly accurate and precise estimation of the cSBP, which was estimated through radial calibration using brachial-cuff SBP and DBP (a type-I tonometric device). Additionally, DCBP proves to be a useful tool for diagnosing central systolic hypertension. It is important to emphasize the need to minimize cuff-blood pressure measurement errors to ensure the reliability of DCBP. Our findings have implications for risk stratification and have the potential to simplify and standardize the evaluation of the incremental value of cSBP beyond brachial SBP. Further research is warranted to explore and validate these findings in larger cohorts and diverse populations.

## Figures and Tables

**Figure 1 jpm-13-01244-f001:**
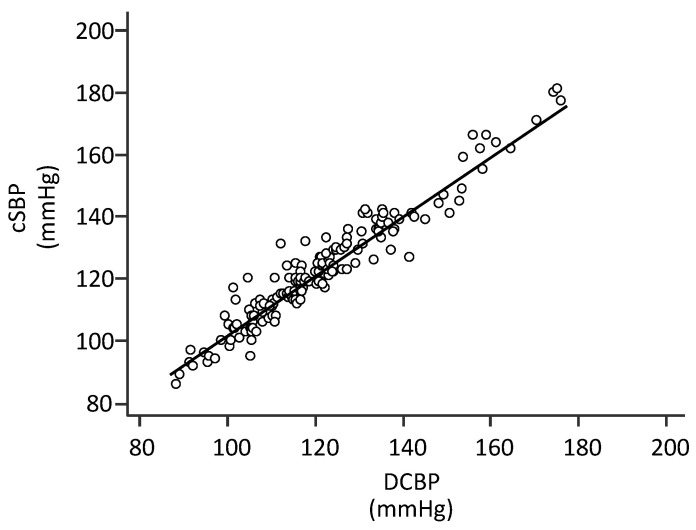
Correlation between DCBP and cSBP. cSBP: central systolic blood pressure estimated by radial applanation tonometry with SphygmoCor (type-I device). DCBP: Direct Central Blood Pressure estimation of cSBP. The equation line is as follows: cSBP = (0.98 × DCBP) + 4 mmHg (*n* = 160; R^2^ = 0.93, *p* < 0.001).

**Figure 2 jpm-13-01244-f002:**
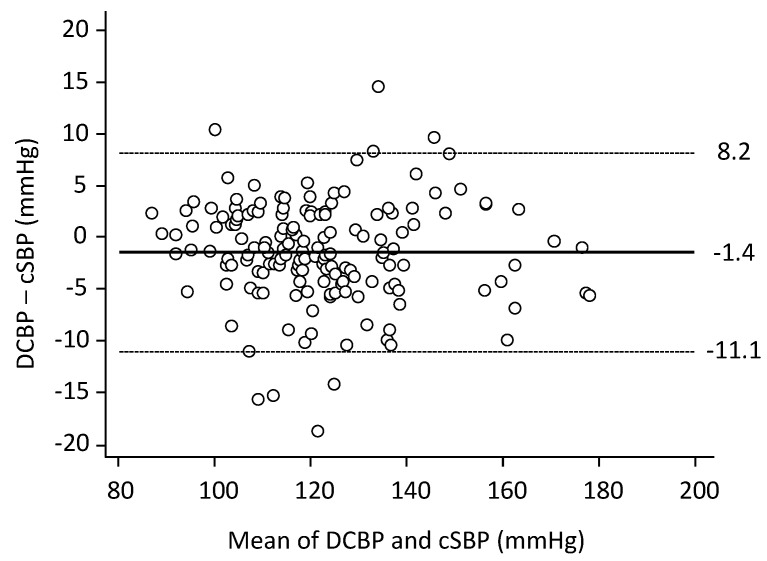
Bland & Altman plots. The (DCBP-cSBP) error was −1.4 ± 4.9 mmHg. Dotted lines indicate 95% CI. There was no influence of the mean on the error (*n* = 160; R^2^ = 0.01; *p* = 0.52).

**Table 1 jpm-13-01244-t001:** Characteristics of the study population (*n* = 160).

Variables	Mean ± SD
Age, years	58 ± 15
Men, *n* (%)	87 (54)
Weight, kg	76 ± 16
Height, cm	168 ± 10
Waist circumference, cm	94.9 ± 13.8
Body mass index, kg/m^2^ (*n* = 158)	27.2 ± 5.0
Normal weight, *n* (%)	57 (36.1)
Overweight, *n* (%)	56 (35.4)
Obese, *n* (%)	45 (28.5)
Clinical and biological data (*n* = 133) ^1^	
Smoker, *n* (%)	44 (33)
Antihypertensive therapy, *n* (%)	105 (79)
Coronary heart disease, *n* (%)	14 (11)
Diabetes mellitus, *n* (%)	30 (22)
Stroke, *n* (%)	2 (2)
Serum creatinine, mmol/L	86.0 ± 29.7

^1^ Clinical and biological information were available in a subset of the whole included population.

**Table 2 jpm-13-01244-t002:** Hemodynamic results (*n* = 160).

Variables	Mean ± SD	Range
Heart rate, beats/min	68 ± 13	45–116
Brachial SBP, mmHg	136.4 ± 18.3	99–211
Brachial DBP, mmHg	80.7 ± 10.4	61–110
Brachial PP, mmHg	55.1 ± 13.3	33–101
Radial MBP, mmHg	98.6 ± 12.6	74–137
cSBP (SphygmoCor), mmHg	123.1 ± 18.3	86.0–181.0
DCBP, mmHg	121.6 ± 18.1	88.3–176.0
Error, mmHg	−1.4 ± 4.9	−18.5–14.5
Error, %	−1.1 ± 3.9	−14.3–11.4
Pulse Wave Velocity, m/s	12.1 ± 3.2	6.9–23.1

cSBP: central systolic blood pressure. DBP: diastolic BP. DCBP: Direct central Blood Pressure estimation. MBP: radial mean BP. PP: pulse pressure. SBP: systolic blood pressure. DCBP was calculated as the MBP^2^/DBP ratio. MBP was the integral of the radial pulse waveform recorded by the tonometer divided by heart period. Error = DCBP-cSBP difference, expressed in mmHg or as a percentage of cSBP.

**Table 3 jpm-13-01244-t003:** Concordance of DCBP to discriminate a 130 mmHg cSBP threshold.

	Sphygmocor cSBP < 130 mmHg	Sphygmocor cSBP ≥ 130 mmHg	Total % of Correctly Classified and Chi-sq Test
DCBP < 130 mmHg	110	8	93% *p* < 0.0001
DCBP ≥ 130 mmHg	3	39	

cSBP: central systolic blood pressure. DCBP: Direct Central Blood Pressure estimation of cSBP using the MBP^2^/DBP formula, where MBP is the integral of the radial pulse waveform recorded by the tonometer divided by heart period, and DBP is brachial diastolic blood pressure (oscillometry).

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
