# Peer review of "Non-Invasive Estimation of Central Systolic Blood Pressure by Radial Tonometry: A Simplified Approach"

_jpm, 2023, doi:10.3390/jpm13081244_

Round 1
Reviewer 1 Report
Chemla and colleagues propose an interesting study comparing an easy-to-perform estimation of central blood pressure with the widespread tonometry-based central blood pressure estimation (through the SphigmoCor® device).
The issue is academically interesting and of clinical relevance. The simplicity of obtaining the proposed estimation makes the DCBP appealing for use in everyday clinical life, including busy clinics.
The study design is appropriate with respect to the study question, the analysis is adequate and appears to have been well-conducted. Importantly, it deserves a praise the use of Bland-Altman analysis and not just correlation analysis [Bland JM, Altman DG. Agreed statistics: measurement method comparison. Anesthesiology. 2012;116(1):182-185].
The main criticism may be that Authors already evaluated their proposed method against the gold standard (i.e. invasive measurement), so that this comparison with the applanation-tonometry meaurements might even appear not to be needed. However, given the extensive use of tonometry-based measurements in several publications, there is a scientific and pragmatic interest in verifying whether the two techniques are comparable. Indeed, only after demonstration of agreement, studies performed using the two techniques will be comparable between them. Authors proved good agreement, so that this study now allows to design future studies using DCBP (with a good confidence that the results might be compared with previous studies).
The paper is globally well-structured and written in a good English. I only have a couple of suggestions, which I have listed below. I feel that its appeal may be increased if Authors would be able to shorten it by approximately 10% of its current length.
The literature is well-done, I just have one suggestion, which is detailed below.
Following specific comments and requests should be addressed and I hope they will help the Authors in further improving the quality of their manuscript.
Lines 25-27: please invert, i.e. state that you compared DCBP (i.e. the new, potentially better / more feasible/easier to implement in clinical practice technique) with cSBP (the already widespread technique). Concretely, please modify into: “In this study, we compared central systolic blood pressure calculated using a Direct Central Blood Pressure estimation (DCBP=MBP2/DBP) with cSBP estimated by radial tonometry.”
Lines 50-52: this is true, but there is much more: it is not only “geographical proximity” to the heart, but also applied physics and physiology. In fact, as Authors for sure already know, there are two important aspects, which should be stressed and clarified to the readers. On one side, systemic vascular resistance is determined mostly by the peripheral arteries (and this manifests mainly in the mean arterial pressure). On the other side, pressure fluctuations (as it can be measured with pulse pressure) in the circulation are determined by arterial compliance, that is mostly influenced by the aorta and the large arteries [Pusterla L et al. Impact of Cardiovascular Risk Factors on Arterial Stiffness in a Countryside Area of Switzerland: Insights from the Swiss Longitudinal Cohort Study. Cardiol Ther. 2022;11(4):545-557. doi: 10.1007/s40119-022-00280-8.]. I think it may be worth to insert a couple of sentences (2-3 lines) clarifying this point to the readership.
Lines 78-79: please verify with your institution, but I feel that you should either write “Diagnostic and Therapeutic Center” (probably better) or “Diagnosis and Therapy Center” (currently, “Diagnosis and Therapeutic Center” is a bit inconsistent: a noun and an adjective).
Line 93, “gender”: this is suboptimal. To investigate the biological effect of sex on arterial compliance, biological sex (i.e., “sex”) should have been preferred over perceived sexual identify (“gender”). Please acknowledge this limitation in your Discussion section.
Line 98: “smoking habits” -> “smoking”
Line 102 (“were retrospectively assessed”): how? Patient chart interrogation, patient-reported (i.e. self-reported) information while filling-in the form, patient recall, ...? Please specify.
Line 105 (“fasting glucose level ≥ 126 mmol/l”): that is an impressive, incredibly high value (probably not compatible with life). Are you sure it was not defined differently? Did you probably rather mean ≥126mg/dL? In this case, I would suggest to use SI units and provide this information as ≥7.0 mmol/L.
Lines 129-130: “using the thumb of rule” -> “using the rule of thumb”
Line 153: “and the ² test was used”: do you mean “and the ?² test was used”? (a symbol has probably disappeared from the manuscript while uploading the file on the submission system, I guess…)
Lines 156-157 (“cardiovascular checkup at the one-day hospital”): do you mean “at the day hospital”, right? If yes, please correct.
Lines 158-160 (“Anamnestic and biological data were available for only 97 patients, and the results are presented for this subgroup.”): Was this not a prospective study (line 77: “enrolled 186 consecutive patients”)? Missing anamnestic and biological data in 63 out of 160 patients would represent a major protocol deviation and should be addressed in the discussion. Why did this happen? Why could this not be prevented? Was any action taken to address this issue during the course of the study?
Furthermore, there is currently no sample size calculation. Can you provide one please? This should show sufficient power also with the "reduced” sample of n=97 participants. Otherwise you must discuss that this study is just explorative and further studies will be needed in order to draw definitive conclusions.
Line 190: “in discriminating discriminate a” -> “in discriminating a”
Line 224: “The FF represents” -> “The form factor (FF) represents”
Line 228: “a single FF” -> “a single form factor”
Lines 273, 275: “FF” -> “form factor”
Line 277 (“By relying on group averages”): what do you mean? Which groups? Why not individually measured values from each patient (even more relevant whilst considering that you are submitting your manuscript to a journal called “J Pers Med”…)?
Line 280, Discussion (general comment)
Another issue, which should briefly be addressed in the Discussion, is the need to generate normative values for different population groups. For example, I am not aware of any normative values of central SBP for children of different ages.
Please see above.
Author Response
Comments and Suggestions for Authors
Chemla and colleagues propose an interesting study comparing an easy-to-perform estimation of central blood pressure with the widespread tonometry-based central blood pressure estimation (through the SphigmoCor® device).
The issue is academically interesting and of clinical relevance. The simplicity of obtaining the proposed estimation makes the DCBP appealing for use in everyday clinical life, including busy clinics.
The study design is appropriate with respect to the study question, the analysis is adequate and appears to have been well-conducted. Importantly, it deserves a praise the use of Bland-Altman analysis and not just correlation analysis [Bland JM, Altman DG. Agreed statistics: measurement method comparison. Anesthesiology. 2012;116(1):182-185].
The main criticism may be that Authors already evaluated their proposed method against the gold standard (i.e. invasive measurement), so that this comparison with the applanation-tonometry meaurements might even appear not to be needed. However, given the extensive use of tonometry-based measurements in several publications, there is a scientific and pragmatic interest in verifying whether the two techniques are comparable. Indeed, only after demonstration of agreement, studies performed using the two techniques will be comparable between them. Authors proved good agreement, so that this study now allows to design future studies using DCBP (with a good confidence that the results might be compared with previous studies).
The paper is globally well-structured and written in a good English. I only have a couple of suggestions, which I have listed below. I feel that its appeal may be increased if Authors would be able to shorten it by approximately 10% of its current length.
The literature is well-done, I just have one suggestion, which is detailed below.
We thank the Reviewer for valuable comments and for appreciating our study.
Following specific comments and requests should be addressed and I hope they will help the Authors in further improving the quality of their manuscript.
- Lines 25-27: please invert, i.e. state that you compared DCBP (i.e. the new, potentially better / more feasible/easier to implement in clinical practice technique) with cSBP (the already widespread technique). Concretely, please modify into: “In this study, we compared central systolic blood pressure calculated using a Direct Central Blood Pressure estimation (DCBP=MBP2/DBP) with cSBP estimated by radial tonometry.”
Thank you. We modified the abstract accordingly.
- Lines 50-52: this is true, but there is much more: it is not only “geographical proximity” to the heart, but also applied physics and physiology. In fact, as Authors for sure already know, there are two important aspects, which should be stressed and clarified to the readers. On one side, systemic vascular resistance is determined mostly by the peripheral arteries (and this manifests mainly in the mean arterial pressure). On the other side, pressure fluctuations (as it can be measured with pulse pressure) in the circulation are determined by arterial compliance, that is mostly influenced by the aorta and the large arteries [Pusterla L et al. Impact of Cardiovascular Risk Factors on Arterial Stiffness in a Countryside Area of Switzerland: Insights from the Swiss Longitudinal Cohort Study. Cardiol Ther. 2022;11(4):545-557. doi: 10.1007/s40119-022-00280-8.]. I think it may be worth to insert a couple of sentences (2-3 lines) clarifying this point to the readership.
Thank you for your comment. We agree that this information shall help the reader. We added a comprehensive sentence and two corresponding references.
- Lines 78-79: please verify with your institution, but I feel that you should either write “Diagnostic and Therapeutic Center” (probably better) or “Diagnosis and Therapy Center” (currently, “Diagnosis and Therapeutic Center” is a bit inconsistent: a noun and an adjective).
Thank you.
- Line 93, “gender”: this is suboptimal. To investigate the biological effect of sex on arterial compliance, biological sex (i.e., “sex”) should have been preferred over perceived sexual identify (“gender”). Please acknowledge this limitation in your Discussion section.
Thank you. Indeed, we misconceived the term. We actually used the biological sex, and we corrected the text.
- Line 98: “smoking habits” -> “smoking”
Done.
- Line 102 (“were retrospectively assessed”): how? Patient chart interrogation, patient-reported (i.e. self-reported) information while filling-in the form, patient recall, ...? Please specify.
Thank you. We modified the sentence as follows:
“…were retrospectively assessed both by scrutinizing the patients documents and by patients interviewing”
- Line 105 (“fasting glucose level ≥ 126 mmol/l”): that is an impressive, incredibly high value (probably not compatible with life). Are you sure it was not defined differently? Did you probably rather mean ≥126mg/dL? In this case, I would suggest to use SI units and provide this information as ≥7.0 mmol/L.
Thank you. We are sorry for the mistake, which we corrected as suggested.
- Lines 129-130: “using the thumb of rule” -> “using the rule of thumb”
Done.
- Line 153: “and the ² test was used”: do you mean “and the ?² test was used”? (a symbol has probably disappeared from the manuscript while uploading the file on the submission system, I guess…)
Thank you. We changed it for “Chi-square”.
- Lines 156-157 (“cardiovascular checkup at the one-day hospital”): do you mean “at the day hospital”, right? If yes, please correct.
Thank you. That was “day hospital”.
- Lines 158-160 (“Anamnestic and biological data were available for only 97 patients, and the results are presented for this subgroup.”): Was this not a prospective study (line 77: “enrolled 186 consecutive patients”)? Missing anamnestic and biological data in 63 out of 160 patients would represent a major protocol deviation and should be addressed in the discussion. Why did this happen? Why could this not be prevented? Was any action taken to address this issue during the course of the study?
Thank you for this important comment. We are aware of this inconvenient. Indeed, the dataset is as old as more than 10 years. We found that some patients were missing due to a dataset error. Finally, we got clinical data from 133 out of 160 included patients. We estimate that this number is safe for clinical interpretation. Furthermore, we compared the characteristics of the included and excluded population and found that they are very similar. Therefore, we exclude the risk of introducing bias. Here we show the statistics:
|
Final population (n=160) |
Excluded population (n=27) |
||||
|
mean |
sd |
mean |
sd |
p value |
|
|
Systolic BP |
136.5 |
18.6 |
136.3 |
16.7 |
0.968 |
|
Diastolic BP |
80.6 |
10.4 |
80.9 |
10.4 |
0.885 |
|
Heart rate |
68.1 |
12.7 |
68.4 |
12.3 |
0.912 |
|
BMI |
27.2 |
5.1 |
27.3 |
4.6 |
0.950 |
|
AGE |
58.6 |
14.8 |
55.0 |
14.3 |
0.265 |
|
Waist/hip ratio |
0.9 |
0.1 |
0.9 |
0.1 |
0.344 |
- Furthermore, there is currently no sample size calculation. Can you provide one please? This should show sufficient power also with the "reduced” sample of n=97 participants. Otherwise you must discuss that this study is just explorative and further studies will be needed in order to draw definitive conclusions.
Thank you. The study was intended as a practical observational prospective trial, with no sample size calculation. Data were analyzed retrospectively for the present manuscript. Anyway, we performed the power calculation, which retrieved values close to 1 both for the correlation analysis and for the Chi-square test. We added this information in the text (Statistical analysis).
- Line 190: “in discriminating discriminate a” -> “in discriminating a”
Done.
- Line 224: “The FF represents” -> “The form factor (FF) represents”
That acronym was expanded three lines before.
- Line 228: “a single FF” -> “a single form factor”
- Lines 273, 275: “FF” -> “form factor”
The acronym FF is broadly used for “form factor”. We do not see why not using it through the text. If the Reviewer does not think that it would it be somewhat misleading, we prefer to continue to use it.
- Line 277 (“By relying on group averages”): what do you mean? Which groups? Why not individually measured values from each patient (even more relevant whilst considering that you are submitting your manuscript to a journal called “J Pers Med”…)?
Thank you. We acknowledge that the sentence was not clear. We therefore rephrased it as follows:
“DCBP can be calculated on existing data, relying on group averages, and providing accurate cSBP estimations, thereby expanding the potential inclusion of patients in research studies or clinical settings.”
- Line 280, Discussion (general comment)
Another issue, which should briefly be addressed in the Discussion, is the need to generate normative values for different population groups. For example, I am not aware of any normative values of central SBP for children of different ages.
Thank you for this important comment. Indeed, the reference values of central blood pressure are lacking worldwide both for children and for adults. Therefore, we do agree with the need to generate normative values, and we modified accordingly the text, as follows:
“An important issue to consider is that few studies analyzing the reference values of cSBP are available. The lack of large-scale worldwide accepted reference values for cSBP should be addressed in future research. At the same time, normative values for DCBP should be calculated or derived to implement the formula in population settings.”
We hope the manuscript has been improved, in line of your comments.
Reviewer 2 Report
1) The authors could provide the area under the ROC curve (ROC analysis) for DCBP in diagnosing a tenometically-derived cSBP of ≥130 mmHg. The Youden index is another approach to assess the diagnostic accuracy of DCBP.
2) The authors could perform some sub-group analyses to demonstrate the robustness of their results. The patients could be stratified into subgroups according to age, gender and the levels of PWV.
3) The authors report that their results may have clinical implications in terms of improved risk stratification. Is the DCBP prognostically informative? Is there any plan to evaluate the prognostic association of DCBP with the risk of cardiovascular morbidity and mortality?
Author Response
The authors thank the reviewer for his/her comments, all of which have been taken into account as follows.
- The authors could provide the area under the ROC curve (ROC analysis) for DCBP in diagnosing a tonometically-derived cSBP of ≥130 mmHg. The Youden index is another approach to assess the diagnostic accuracy of DCBP.
This has been done.
-it is now indicated in the abstract and results: “A DCBP value > 126 mmHg exhibited a sensitivity of 91.5% and specificity of 94.7% (Youden index = 0.86) in discriminating a cSBP threshold of 130 mmHg.”
- it is now indicated in the statistical analysis section: “ Finally, ROC curve analysis was performed to test the ability of DCBP in confirming the diagnosis of central systolic hypertension.”
- the corresponding ROC curve is now presented as new Figure 3
- The authors could perform some sub-group analyses to demonstrate the robustness of their results. The patients could be stratified into subgroups according to age, gender and the levels of PWV.
This has been done and a new paragraph detailing the results has been included in the results section. It is also indicated that “In all the subgroups defined according to gender, age or PWV, the (DCBP – cSBP) mean error was ≤5mmHg and the SD was ≤8mmHg.”
- The authors report that their results may have clinical implications in terms of improved risk stratification. Is the DCBP prognostically informative? Is there any plan to evaluate the prognostic association of DCBP with the risk of cardiovascular morbidity and mortality.
It is now indicated at the end of the Limitations section that: “This is a validation (non-invasive) study, and it was not designed to test whether DCBP was prognostically informative”.
We indeed plan to evaluate the prognostic association of DCBP with the risk of cardiovascular morbidity and mortality, but this is too preliminary to be included in the present manuscript, in our opinion. We also hope the manuscript will stimulates research in that direction. It is now indicated at the end of the Implications section: “Finally, our study indicates that DCBP may be useful in diagnosing central systolic hypertension, and further studies are needed to confirm this finding and to document the potential prognostic association of DCBP with cardiovascular risk.”
We hope the manuscript has been improved, in line with your comments.